

# Vertically resolved aerosol variability at the Amazon Tall Tower Observatory under wet season conditions

Marco A. Franco[1,2,a], Rafael Valiati[1], Bruna A. Holanda[2,b], Bruno B. Meller[1,2], Leslie A. Kremper[2], Luciana V. Rizzo[1], Samara Carbone[3], Fernando G. Morais[1], Janaína P. Nascimento[1,c], Meinrat O. Andreae[2,4,5], Micael A. Cecchini[6], Luiz A. T. Machado[1,2], Milena Ponczek[1], Ulrich Pöschl[2], David Walter[2,7,d], Christopher Pöhlker[2], and Paulo Artaxo[1]

[1] Institute of Physics, University of São Paulo, São Paulo 05508-900, Brazil

[2] Multiphase Chemistry Department, Max Planck Institute for Chemistry, 55128 Mainz, Germany

[3] Federal University of Uberlândia, Uberlândia-MG, 38408-100, Brazil

[4] Scripps Institution of Oceanography, University of California San Diego, La Jolla, CA 92093, USA

[5] Department of Geology and Geophysics, King Saud University, Riyadh, Saudi Arabia

[6] Department of Atmospheric Sciences, Institute of Astronomy, Geophysics and Atmospheric Sciences, University of São Paulo, Rua do Matão, 1226, Cidade Universitária, 05508-090, Brazil

[7] Department of Biogeochemical Systems, Max Planck Institute for Biogeochemistry, 07701 Jena, Germany

[a] now at Department of Atmospheric Sciences, Institute of Astronomy, Geophysics and Atmospheric Sciences, University of São Paulo, Rua do Matão, 1226, Cidade Universitária, 05508-090, Brazil.

[b] now at Hessian Agency for Nature Conservation, Environment and Geology, 65203 Wiesbaden, Germany

[c] now at NOAA ESRL Global Systems Laboratory, Boulder, United States

[d] now at Climate Geochemistry Department, Max Planck Institute for Chemistry, 55128 Mainz, Germany

**Correspondence:** Marco A. Franco (marco.franco@usp.br), Christopher Pöhlker (c.pohlker@mpic.de)

**Abstract.** The wet season atmosphere in the central Amazon resembles natural conditions with minimal anthropogenic influence, making it one of the rare pre-industrial-like continental areas worldwide. Previous long-term studies have analyzed the properties and sources of the natural Amazonian background aerosol. However, the vertical profile of the planetary boundary layer (PBL) has not been assessed systematically. Since 2017, such a profile assessment has been possible with the 325 m

high tower at the Amazon Tall Tower Observatory (ATTO), located in a largely untouched primary forest in central Amazonia. This study investigates the variability of submicrometer aerosol concentration, size distribution, and optical properties at 60 and 325 m height in the Amazon PBL. The results show significant differences in aerosol volumes and scattering coefficients in the vertical gradient. The aerosol population was well-mixed throughout the boundary layer during the daytime but became separated upon stratification during nighttime. We also found a significant difference in the spectral dependence of

the scattering coefficients between the two heights. The analysis of rainfall and related downdrafts revealed changes in the aerosol populations before and after rain events, with absorption and scattering coefficients decreasing as optically active particles are removed by wet deposition. The recovery of absorption and scattering coefficients is faster at 325 m than at 60 m. Convective events were concomitant with rapid increases in the concentration of sub-50 nm particles, likely associated with downdrafts. We found that the aerosol population near the canopy had a significantly higher mass scattering efficiency than at

325 m. It was also observed a clear spectral dependence, with values for $\lambda$ = 450, 525 and 635 nm of 7.74±0.12, 5.49±0.11 and 4.15±0.11 m$^2$ g$^{-1}$, respectively, at 60 m, while at 325 m, the values were 5.26±0.06, 3.76±0.05 and 2.46±0.04 m$^2$ g$^{-1}$,





respectively. The equivalent aerosol refractive index results, which were obtained for the first time for the wet season in central Amazon, show a slightly higher scattering (real) component at $60\,\mathrm{m}$ compared to $325\,\mathrm{m}$, of 1.33 and 1.27, respectively. In contrast, the refractive index's absorptive (imaginary) component was identical for both heights, at 0.006. This study shows that the aerosol physical properties at 60 and $325\,\mathrm{m}$ height are different, likely due to aging processes, and strongly depend on the photochemistry, PBL dynamics, and aerosol sources. These findings provide valuable insights into the impact of aerosols on climate and radiative balance and can be used to improve the representation of aerosols in global climate models.

## 1 Introduction

The central Amazon is one of the few continental places where the atmospheric aerosol population, along with aerosol-cloud interactions, approaches conditions without significant anthropogenic impacts during particular periods of the wet season (February to May) (Martin et al., 2010; Pöhlker et al., 2016, 2018; Andreae et al., 2022). Cloud and precipitation formation are directly linked to the concentration and properties of atmospheric aerosols, which greatly affects the hydrological cycle of the basin (Andreae et al., 2001; Andreae, 2007; Artaxo et al., 2022; Khadir et al., 2023). In the Amazon, biogenic emissions, such as organic gases and aerosols, combined with the high concentration of water vapor and intense solar radiation, result in an atmosphere particularly susceptible to different external forcings that may change its composition (Pöhlker et al., 2012; Carslaw et al., 2013; Artaxo et al., 2022). The episodic occurrence of clean conditions in today's atmosphere is critically important to study the fundamental processes that control the natural aerosol population in the Amazon and beyond (Holanda et al., 2023). This includes, for instance, the diverse sources of primary biological aerosol particles (Pöhlker et al., 2012; Fröhlich-Nowoisky et al., 2016; Šantl-Temkiv et al., 2020; Prass et al., 2021) as well as the likewise diverse emissions of volatile organic compounds (VOC), followed by cascades of atmospheric oxidation that form secondary organic aerosols (SOA) through new particle formation (NPF) and condensation (Andreae et al., 2022; Kirkby et al., 2016; Rose et al., 2018; Edtbauer et al., 2021; Pfannerstill et al., 2021; Yáñez-Serrano et al., 2020).

Earlier studies have documented the physical and chemical properties of the atmospheric aerosol in the central Amazon (e.g., Rizzo et al., 2011; Artaxo et al., 2013; Rizzo et al., 2013; Pöhlker et al., 2016, 2018; Rizzo et al., 2018; Saturno et al., 2018b; Talbot et al., 1988, 1990). At places such as the Amazon Tall Tower Observatory (ATTO) site, the strongly seasonal aerosol optical properties and particle number concentrations were defined by the different aerosol sources and atmospheric aging involved (Pöhlker et al., 2016; Saturno et al., 2018b; Franco et al., 2022). Under very clean atmospheric conditions during the wet season, the submicrometer aerosol population is dominated by sub-50 nm and Aitken mode particles (mode centered on average at $\mathrm{D}_{AIT}\sim70$ nm). Moreover, there is also a significant amount of aerosols of the accumulation mode ($\mathrm{D}_{ACC}\sim160$ nm), mainly from particle growth events and horizontal atmospheric transport. In clean atmospheric conditions like these, the mean total aerosol concentration reaches values as low as $\sim250\ cm^{-3}$ (Pöhlker et al., 2018; Franco et al., 2022; Machado et al., 2021).

During the wet season, long-range transport (LRT) plumes from the African continent often affect atmospheric composition and air quality in central Amazonia (Pöhlker et al., 2016; Talbot et al., 1990; Moran-Zuloaga et al., 2018; Holanda et al., 2023).





These plumes consist mainly of dust and aged smoke from biomass burning but also include marine primary and secondary aerosols. The difference of up to 20 nm in the accumulation mode between local aerosols and LRT plumes suggests that aging significantly occurs in the particles (Pöhlker et al., 2018; Holanda et al., 2020). These LRT events can cause aerosol concentrations to reach levels as high as $430 \, cm^{-3}$, and the chemical composition of these aerosols is dominated by dust,

smoke, sea salt, and marine biogenic sulfate (Andreae et al., 1990; Holanda et al., 2020, 2023). Coarse mode aerosols also exhibit significant variations during LRT events (Moran-Zuloaga et al., 2018).

During the wet season in the Amazon, aerosol optical properties show distinct characteristics compared to the dry season. The average scattering and absorption coefficients during the wet season are 7.5 and 0.68 $Mm^{-1}$, respectively, with a single scattering albedo, $\omega_0$, of 0.93 for clean periods and 0.80 for LRT periods (Saturno et al., 2018b; Holanda et al., 2023). The

high value of $\omega_0$ during the clean periods indicates that these aerosols are efficient in scattering radiation and are likely SOA. In contrast, during LRT events, the plumes originated from the African continent are dominated by dust and aged biomass burning particles (Holanda et al., 2023).

In contrast, the scattering and absorption coefficients during the dry season from August to November are on average 33 and 4 $Mm^{-1}$, respectively, with $\omega_0 \sim 0.87$ (Saturno et al., 2018b). The aerosols are comparatively much more absorbing during

this period. This is due to the increased influence of regional emissions, such as biomass burning and anthropogenic activities, which occur in several Amazon regions (Ponczek et al., 2022) and a decrease in wet deposition. It is worth noting that the aerosol optical properties have important implications for the Earth's radiation budget and climate (Liu et al., 2020). Highly scattering aerosols, such as those present during the wet season in the Amazon, predominantly reflect incoming solar radiation back to space and have a cooling effect on the climate. On the other hand, more absorbing aerosols, such as those present

during the dry season, trap radiation in the atmosphere and lead to warming (Boucher, 2015).

Previous *in-situ* studies on long-term optical and physical properties of aerosols in the central Amazon have relied on measurements taken at a single height (Rizzo et al., 2011, 2013; Artaxo et al., 2013; Saturno et al., 2018a). For instance, at the ATTO site, measurements were obtained at the height of 60 m, which is relatively close to the canopy top located at about ~40 m height. However, these studies have not investigated the vertical variability of aerosols throughout the boundary layer

in the region. This information is crucial for understanding the vertical distribution of aerosols and their sources, transport, and fate, which are essential for accurate modeling and predictions of aerosol impacts on climate and health.

With its 325-m tall tower, the ATTO site offers a unique opportunity to investigate aerosol physical properties at different heights. Here, we use 60 and 325 m heights, the locations of the aerosol inlet lines at ATTO (Andreae et al., 2015; Machado et al., 2021). By measuring aerosol properties at different heights, it is possible to better understand the vertical distribution of

aerosols and the processes that influence them and characterize the temporal variability of the vertical gradients. For example, measuring at multiple heights can help distinguish between local and regional sources of aerosols and provide insights into the role of atmospheric transport and mixing in shaping the aerosol distribution. Additionally, the data collected at different heights can be used to validate and improve aerosol models and remote sensing retrievals. Characterizing aerosol properties at two or more heights is an important step toward comprehensively understanding aerosol sources, transport, and impacts in the



Amazon and other regions. So far, there are only two bioaerosol studies that resolved the profiles (Prass et al., 2021; Barbosa et al., 2022), but none of them focused on the aerosol fine mode systematically.

This study aims to improve our understanding of the vertical distribution of atmospheric aerosols in the lower Amazonian troposphere. Specifically, we investigate the variability of optical properties, particle number concentration, and size distribution of aerosols at two different heights (60 and 325 m) at the ATTO site. By characterizing the vertical distribution of aerosols, we shed light on the sources, transport, and fate of these particles and improve our understanding and ability to predict their impacts on biosphere-atmosphere exchange and climate. Comparing results at different heights provides insights into the processes that shape aerosol distribution and helps validate and improve aerosol models and remote sensing retrievals. Overall, this study has important implications for our understanding of aerosol impacts on the Amazonian ecosystem.

## 2 Measurements and data analysis

### 2.1 The Amazon Tall Tower Observatory

This study's primary data was obtained from the Amazon Tall Tower Observatory (ATTO), a remote scientific station located in the central Amazon, about 150 km northeast of Manaus (S 02° 08.9', W 59° 00.2', 120 m.a.s.l.), comprised of two 80-m high towers, on one of which the 60 m inlet is located, and the 325-m tall ATTO tower, about 650 m apart. The ATTO site is thoroughly described in Andreae et al. (2015), providing essential details about the site's location and setting.

### 2.2 Terminology

The seasonal classification adopted in this study follows Pöhlker et al. (2016), where the wet season is defined as February to May, and the dry season spans from August to November. The transition period between the two seasons is denoted by June and July, while December and January mark the transition between the dry and wet seasons. To describe aerosol size distributions, we employed the widely recognized nomenclature for multi-modal classification (Boucher, 2015; Seinfeld and Pandis, 2012). Specifically, particles with diameters greater than 100 nm and smaller than 400 nm are classified as the accumulation mode, while the Aitken mode comprises particles ranging from 50 to 100 nm in diameter. The smallest particles were referred to as the "sub-50nm mode," encompassing particles measuring between 10 and 50 nm in diameter, as defined by Franco et al. (2022). Particles larger than 400 nm were not investigated in this study due to the limitations of the SMPS. (Franco et al., 2022; Rizzo et al., 2018).

### 2.3 Aerosol number size distributions

To measure the size distributions of fine mode particles, referred to as Particle Number Size Distributions (PNSDs), we employed two Scanning Mobility Particle Sizers (SMPSs), with sample inlets located at 60 and 325 meters above ground level. The TSI Inc. Model 3082 instruments were coupled with CPCs (Condensation Particle Counter) Model 3772. Air samples were transmitted through metal tubes to the instruments located in air-conditioned containers on the ground beside the towers. Prior



to entering the instruments, the air was dried to 40% relative humidity using different driers, as described elsewhere (Franco et al., 2022; Machado et al., 2021). Additional details on the aerosol transmission system can be found elsewhere (Andreae et al., 2015; Franco et al., 2022). The SMPSs utilized in this study covered an aerosol diameter range from 10 to 400 nm, including 104-size bins. More details can be found in the literature (Machado et al., 2021; Franco et al., 2022). For the wet

season months of 2019 and 2020, the 30-minute average data coverage for both instruments was satisfactory, with 70.5% of the studied period covered by the instrument measuring at 60 m and 91.8% by the one at 325 m.

To ensure consistent measurements, all PNSDs were corrected for standard conditions of temperature and pressure (273.15 K and 1013.25 hPa). Moreover, due to the required connections between the inlet and the instruments on the ground, sampled air had to traverse transmission lines, which led to losses. To compensate for these losses, similarly to what was done in

Holanda et al. (2023), we corrected all particle number size distributions measured at both 60 and 325 m, within the entire measurement range of the instrument (10 to 400 nm), with transmission efficiency curves. These curves were calculated using software developed by von der Weiden et al. (2009), accounting for factors such as geometry, sedimentation, diffusion losses, and electrostatic and turbulent deposition. It is worth noting that the surfaces of the tubes mostly affected the smallest particles (< 50 nm) due to Brownian diffusion. Figure S1 shows the correction curves obtained by the calculations. For the particles

at the end of the Aitken - accumulation mode, which dominate the optical properties, we did not need to correct the optical measurements.

## 2.4  Aerosol optical properties

The optical properties of the aerosol particles were obtained at the ATTO site using a variety of instruments. Nephelometers, specifically Aurora Ecotech 3000 models, were used to measure scattering coefficients at both height levels at three wavelengths

in the visible spectrum: 450, 525, and 635 nm. Truncation corrections were made according to the literature (Müller et al., 2011). The absorption component of the aerosols was measured using MAAPs (Multi-Angle Absorption Photometers) and aethalometers (model AE33) that determine the absorption coefficient across seven wavelengths (370, 470, 520, 590, 637, 880, and 950 nm). The aethalometer data was corrected similarly to Saturno et al. (2018b). As both MAAP and aethalometers measure at 637 nm, a comparison between the two was performed to ensure that there were no calibration issues.

### 2.4.1  Aerosol intensive properties

Measurements of the aerosol optical properties at multiple wavelengths allow estimations of their intensive properties. Specifically, the single scattering albedo (SSA, $\omega_0$) was calculated using Equation 1, where $\sigma_{abs}$ and $\sigma_{scat}$ are the absorption and scattering coefficients, respectively, in units of $\text{Mm}^{-1}$. To calculate the $\omega_0$, the scattering coefficients measured by the nephelometers had to be extrapolated to 637 nm since the nephelometers and MAAPs do not measure at the same wavelengths. By

doing so, we can compare the optical properties at the same wavelength and obtain a valid $\omega_0$ value.

$$\omega_{0,\lambda} = \frac{\sigma_{scat,\lambda}}{\sigma_{ext,\lambda}} = \frac{\sigma_{scat,\lambda}}{\sigma_{abs,\lambda} + \sigma_{scat,\lambda}} \qquad (1)$$



The scattering and absorption Ångström coefficients were estimated through linear regression of Equation 2, where $\lambda$ is the wavelength and å is the mean Ångström exponent, which can be the scattering (SAE) or absorption (AAE) Ångström exponent.

$$ln(\sigma_\lambda) = -\text{å}\ln(\lambda) + C \tag{2}$$

## 2.5 Refractive index estimation

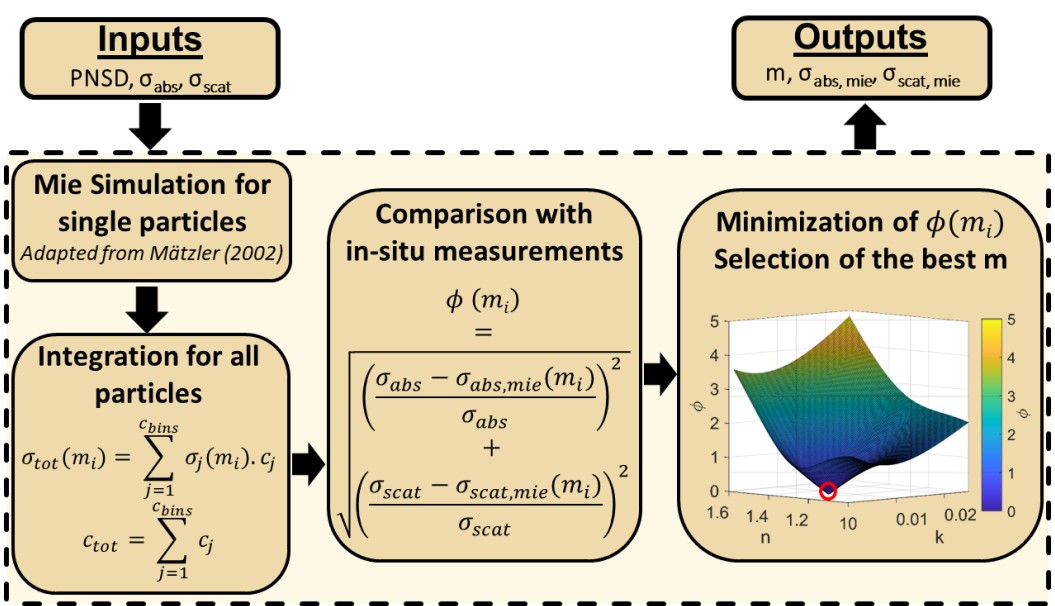

**Figure 1.** Flow diagram of the algorithm used to estimate the refractive index. The algorithm takes the particle number size distribution (PNSD) as well as the scattering and absorption coefficients ($\sigma_{scat}$ and $\sigma_{abs}$, respectively) as input parameters. The algorithm proceeds with the Mie Simulation, which calculates $\sigma_{scat}$ and $\sigma_{abs}$ for each PNSD. The simulated optical parameters are then compared to the *in-situ* measurements by the $\phi$ function. This comparison involves plotting $\phi$ against the refractive index real (**n**) and the imaginary (**k**) parts to determine the optimal parameters that yield the minimum $\phi$.

5    The refractive index ($\mathbf{m}(\lambda)$) is a complex mathematical quantity, a function of the wavelength $\lambda$, used to describe the inter-action between electromagnetic waves and matter. Both real ($\mathbf{n}(\lambda)$) and imaginary ($\mathbf{k}(\lambda)$) components of $\mathbf{m}(\lambda)$ are intensive aerosol properties, i.e., they do not depend on the aerosol concentration but rather on their intrinsic chemical and physical properties. In this study, we use the following definition of the refractive index: $\mathbf{m}(\lambda) = \mathbf{n}(\lambda) - i\mathbf{k}(\lambda)$, similarly to what is found elsewhere (Seinfeld and Pandis, 2012; Boucher, 2015). The $\mathbf{n}(\lambda)$ component is associated with radiation scattering, and the

10   $\mathbf{k}(\lambda)$ component is related to radiation absorption by the aerosols in the atmosphere.

The aerosol refractive index is fundamental to modeling the aerosol's interaction with the incident radiation. It provides information about its radiative impact and efficiency of scattering and absorbing radiation at a particular wavelength. In this



study, we applied the Mätzler collection of Matlab scripts (Jacques, 2010) that use Mie Theory calculations to a homogeneous single spherical particle of known refractive index. Using the core of Mätzler's scripts, we developed an algorithm to estimate the equivalent refractive index of the observed aerosol size distribution as an inversion problem. Figure 1 shows a scheme of the algorithm. It utilizes the particle number size distribution and the scattering and absorption coefficients as input parameters.
The algorithm then proceeds to perform the Mie simulation, which computes $\sigma_{scat}(\lambda)$ and $\sigma_{abs}(\lambda)$) for each specific PNSD.

The simulated optical parameters are subsequently compared with the on-site measurements using the $\phi$ function. This comparative analysis identifies the optimal parameters that produce the minimum $\phi$. This methodology differs from the one used in Rizzo et al. (2013), in particular, because it consumes less computational processing time to estimate the indexes, as Mie calculations are performed once and are not needed for every point in time. Also, it gives more control of the intermediate
variables related to calculating the output.

## 2.6 Period selection

The core study period with simultaneous data coverage at both heights was defined as the months of April and May 2019 and 2020, with notable exceptions. To assure data quality and the selection of the cleanest periods based on Pöhlker et al. (2018) and Franco et al. (2022), the following data configuration was used in this study:

– Nephelometer data: April and May 2019. During this period, the two instruments were calibrated three times to ensure the quality of the comparisons. The scattering coefficient data for 2020 were not used for vertical comparison, as the COVID-19 pandemic did not make it possible to calibrate the instruments with regular frequency and thus guarantee the robustness of the comparisons.

   – MAAP, AE33, SMPS, and CPC data: April and May 2019 and 2020. These instruments do not require frequent calibra-
tions, and their data can be used for vertical comparisons. In particular, simultaneous AE33 data comprising the selected periods of the wet season was only available in May 2020.

   – The two-height meteorological characterization was performed using the entire dataset from 2013 to 2022 for statistical significance. The temperature and humidity conditions in the wet season are very similar among different years, justifying the calculation of interannual averages.

– For the downdraft studies, the selection of months was expanded for the entire wet season (February to May 2019 and 2020) to increase the statistical significance of the results. The 325-m nephelometer was available only in 2019.

## 2.7 Meteorological Parameters

Meteorological parameters, such as air temperature ($T$), pressure ($p$), wind speed and direction, and relative humidity ($RH$), were measured at two different heights at the ATTO site. This study considers the period from 2013 to 2022 as a consistent
database for meteorological characterization. From 2013 to 2018, the measurements were conducted at the 80 m high tower, with instruments installed at about 60 m height above the ground. The wind speed and direction sensors had instrumental





problems during 2017 and 2018, so the analysis did not consider these data. Between 2019 and April 2022, all measurements were conducted using a compact weather station (Lufft, WS600-LMB, G. Lufft Mess- und Regeltechnik GmbH, Fellbach, Germany) installed at 321 m height on the ATTO Tall Tower. The dataset allowed a precise meteorological description of the site and an analysis of the vertical gradient of these parameters. Detailed information on the meteorological instruments can be
found in Andreae et al. (2015) and Franco et al. (2022).

## 2.8 Downdraft detection

To study the influence of meteorological events on the analyzed aerosol properties, we developed a method for detecting downdrafts. We applied it to the considered months (February to May), using each instrument's appropriate periods, as discussed in Section 2.6. The method to detect convective events uses the time series of equivalent potential temperature anomaly ($\Delta\theta$),
which is an adequate tracer for this type of event (Machado et al., 2002; Gerken et al., 2016; Franco et al., 2022). The $\Delta\theta$ was calculated according to Franco et al. (2022). Here, the method was constructed considering the time series derived from a 90-minute moving average of the $\Delta\theta$ value (that is, an average of 3 points, since data is obtained every 30 minutes), which we will call $\Delta\theta_{mov}$, to smooth the curve and decrease the algorithm's sensitivity to short and weak events. From there, we considered that the beginning of a convective event occurs when $\Delta\theta_{mov}$ falls below the value given by the $25^{th}$ percentile
of the entire series ($\Delta\theta_{25}$). This value is usually between -1 and -2 K for different analysis periods, reducing the algorithm's sensitivity by discarding events of very weak intensity. In addition, we also defined that the convective event ends when $\Delta\theta_{mov}$ > $\Delta\theta_{25}$. Figure S2 shows an example of the application of this algorithm for a representative period of 6 days when multiple events were detected.

This algorithm was able to identify 188 intense downdraft events during the wet season months of 2019 and 2020, with an
average duration of 7 hours and mean $\Delta\theta_{min}$ of -5.7 K. Finally, it was possible to select events to eliminate those with higher $\Delta\theta_{min}$ and consider only the most intense events in subsequent analyzes. For example, for the same data period, excluding events with $\Delta\theta_{min}$ > $\Delta\theta_{10}$ (where $\Delta\theta_{10}$ is the value given by the $10^{th}$ percentile of the entire series, usually between -4 and -5 K), we are left with 120 events, with an average duration of 9.5 hours and mean $\Delta\theta_{min}$ of -7.1 K. The comparison shown in Figure S2 indicates the types of events typically discarded by this filter. The analyses presented in Section 3.4 were performed
with the intense event filter on to attenuate the noise caused by weak and short downdrafts and to understand the changes in the properties observed in the boundary layer caused by air parcels moving during strong convective events. Additionally, Figure S11 shows that the most common times to detect the onset of a convective event roughly coincide with precipitation peaks in the early morning and mid-afternoon, which are the periods related to more growth particle events (Franco et al., 2022). Only 32% of events start during the nighttime.



# 3   Results and Discussions

## 3.1   Meteorological conditions

In the period comprised of April and May 2019 and 2020, the average T and RH, measured at 325 m height, were 24.6 °C
and 85.2 %, respectively. In comparison, the average T and RH values at 60 m were 24.9 °C and 82.8 %, respectively. The
average accumulated precipitation between April and May was 685 mm, representing ∼63% of all accumulated precipitation
in the wet season. Figure S3 shows the wind rose for the period studied for 2019 and 2020. On average, the wind speed was
4.4 $m/s$, a value lower than the average over the entire wet season of the same year of 5.3 $m/s$. The dominant wind directions
coincide with the wet season directions obtained by previous studies, i.e., mainly from the northeast (Pöhlker et al., 2019),
which provides air mass origins from untouched Amazonian vegetation.

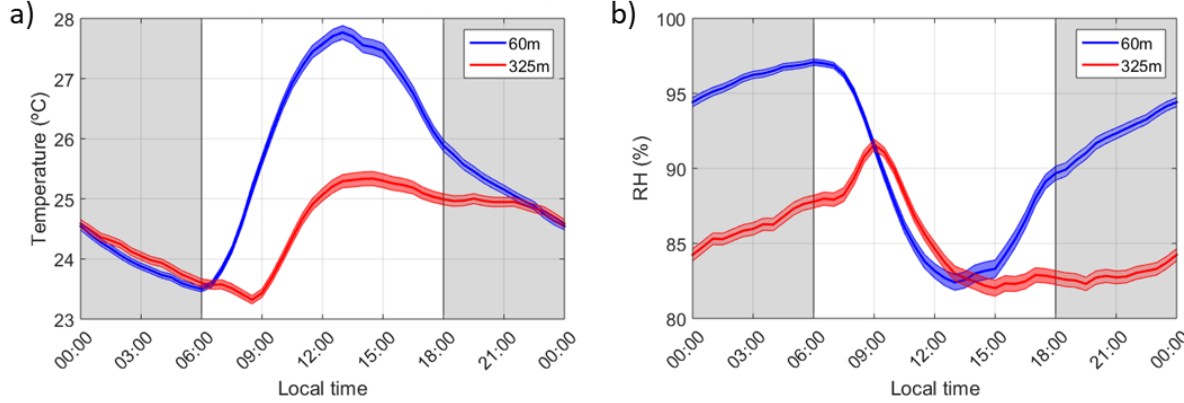

**Figure 2.** Diurnal cycle of a) temperature and b) relative humidity (RH) at two different heights at the ATTO site during April and May.
The blue and red lines represent the averages of the measurements at 60 and 325 m height, respectively, and the shaded areas represent the
standard error of the mean. The 60 m measurements were performed between 2013 and 2018, and the 325 m measurements from 2018 to
2022.

Figure 2 presents the diurnal pattern of the meteorological variables measured at 60 (red curve) and 325 m (blue curve)
during the cleanest months. Although different periods were considered in this particular analysis, the main objective here is to
show the behavior of the diurnal cycle at each height, considering the maximum comprised data period. The temperature during
the early afternoon is, on average, 2 °C higher at 60 than at 325 m. With the development of the nocturnal stable boundary
layer, this temperature profile is reversed at nighttime. The temperature at canopy height begins to rise right after sunrise, while
it takes around three more hours for the morning warming to be observed at 325 m. This behavior is also observed in the
diurnal cycle of relative humidity (Figure 2b). While during most of the day, RH is lower at 325 m, both curves are nearly
identical during the warming hours (from 09 LT to 15 LT). All the results obtained for the 60 m level agree with the site's



meteorological description made by Andreae et al. (2015), who reported the prevalence of ENE winds during the wet season and daily temperatures between 22.5 and 28 °C during these months.

## 3.2  Aerosol vertical gradient

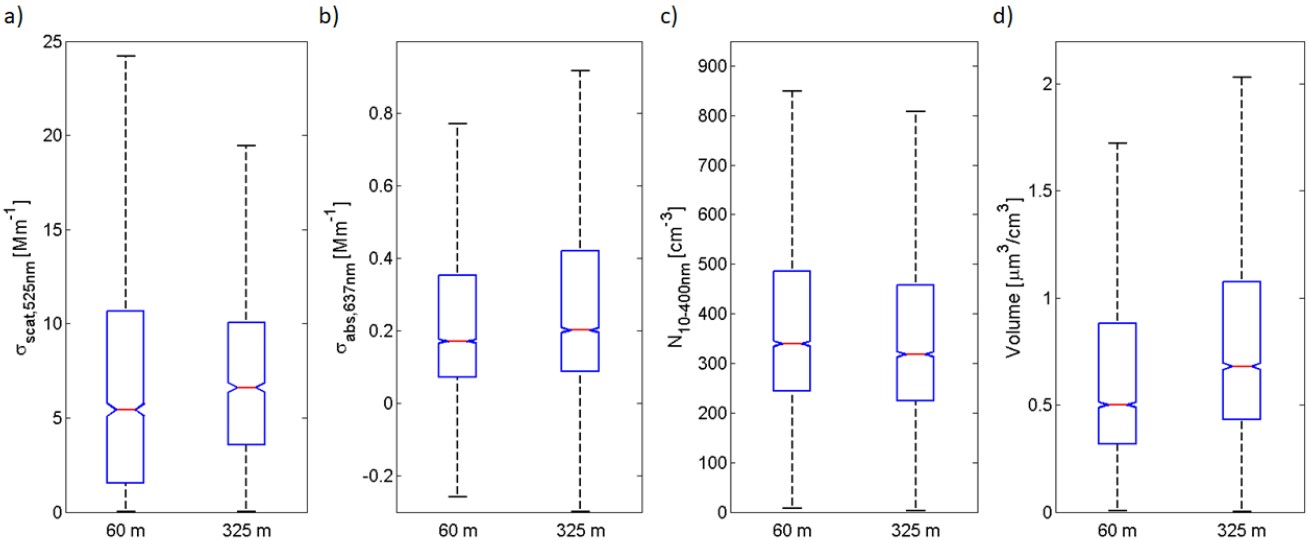

**Figure 3.** Box-and-whisker plots of the optical aerosol properties during the wet seasons of 2019 and 2020, showing a) $\sigma_{scat,525nm}$ and b) $\sigma_{abs,637nm}$ ), c) particle number concentration, and d) aerosol volume at 60 and 325 m height. The boxes represent the quartiles, the whiskers represent the 90th and 10th percentiles, and the horizontal line represents the median.

Figure 3 presents direct comparisons between optical properties, particle number concentration, and fine mode aerosol vol-
ume in the vertical profile, with simultaneous measurements at 60 and 325 m for a given instrument. The scattering coefficient (Figure 3a), $\sigma_{scat,525nm}$, at 60 m height had a median and interquartile range (IQR) of 5.4 (1.5 - 10.1) Mm$^{-1}$, while at 325 m height, the median and IQR were 6.6 (3.6 - 10.0) Mm$^{-1}$. The statistical significance of this difference was verified through the Wilcoxon rank-sum test, which returned a p-value < 0.01, indicating that differences in scattering coefficients are significant at a significance level of 99%. Figures S4a and b show the spectral dependence of the aerosol scattering coefficient at 60 and
325 m height, respectively. At all wavelengths, the median scattering coefficients are higher at 325 m, with medians of 8.6, 6.6, and 5.1 Mm$^{-1}$ for $\lambda = 450$, 525, and 635 nm, respectively. At 60 m, the medians of the scattering coefficients are 6.8, 5.4, and 4.6 Mm$^{-1}$ for $\lambda = 450$, 525, and 635 nm, respectively. These differences are directly reflected in the scattering Ångström exponent (Figure S4c), where the median of this coefficient at 60 m is 1.29, while at 325 m, it is 1.61.

Note that the SAE is an intensive property that does not depend directly on the aerosol number concentration and shows
that aerosol populations at different heights have different spectral dependencies on incident radiation, particularly with higher scattering coefficients at 325 m. In particular, the SAE differences may be linked to the influence of different coarse mode



fraction aerosol populations at 60 and 325 m. The lower SAE value has been reported to represent aerosol populations with size distributions dominated by larger particles (Schuster et al., 2006). Prass et al. (2021) have shown that the bioaerosols have a different population distribution along with the vertical profile of the ATTO tower. For instance, the study reported different populations of eukaryotic and bacterial particles between 60 and 325 m, a significant component of the coarse aerosol mode in central Amazon. Thus, biogenic aerosols may play a significant role in aerosol optical properties, which has important implications for radiative balance.

Figure 3b compares the absorption coefficients, $\sigma_{abs,637nm}$. At 60 m height, the coefficient at 637 nm had a median and IQR of 0.17 (0.07 - 0.35) Mm$^{-1}$, while at 325 m, the median and IQR were 0.20 (0.09 - 0.42) Mm$^{-1}$. The statistical significance obtained with the Wilcoxon rank-sum test for the absorption coefficient also presented a p-value < 0.01, indicating that the differences are statistically significant. However, the relatively small difference could also be associated with the instrumental uncertainties due to the very low absorption coefficients measured in the cleanest months of the wet season, so $\sigma_{abs,637nm}$ is approximately the same at both heights. Figures S5a and b show the spectral dependence of the absorption of the coefficients measured at 60 and 325 m, calculated from the AE33 data for May 2020 and corroborate the observation that the absorption coefficients are slightly higher at 325 than at 60 m. Specifically, at 325 m, the medians of the coefficient varied between 0.63 and 0.20 Mm$^{-1}$ for $\lambda = 370$ and 950 nm, respectively. At 60 m, the medians varied between 0.60 and 0.18 Mm$^{-1}$ for $\lambda = 370$ and 950 nm, respectively. Furthermore, the similarity between the absorbing component of aerosols is reflected in the Ångström absorption exponent, $\mathring{a}_{Abs}$, whose medians at 60 and 325 m are, respectively, 1.27 and 1.24. The single scattering albedo, $\omega_{0,637nm}$, also an intensive aerosol property, is identical for populations at both height levels. The median $\omega_{0,637nm}$ and its IQR at 60 m were 0.94 (0.90 - 0.96), while at 325 m, the median $\omega_{0,637nm}$ and its IQR were 0.94 (0.92 - 0.95). Although differences were observed for the SAE and the vertical profile, indicating different aerosol coarse mode populations, the $\omega_{0,637nm}$ indicates that the fine mode aerosol population in both highs is very efficient in scattering radiation.

Figure 3c compares aerosol number concentrations in the vertical profile within the 10-400 nm size range measured by the SMPS. The results showed similar medians at 60 m and 325 m: at 60 m, the median and IQR of $N_{10-400}$ were 340 (240-490) $cm^{-3}$, while at 325 m, the median and IQR of $N_{10-400}$ were 320 (220-460) $cm^{-3}$. While the differences in the median values of the number concentrations were less than 10%, which also could be related to instrument counting efficiencies for particles smaller than 50 nm, the fine mode aerosol volume (Figure 3d) showed significant differences between the two heights. At 60 m, the median volume and IQR were 0.50 (0.32 - 0.88) $\mu m^3\ cm^{-3}$. At 325 m, the median and IQR of the volume were 0.68 (0.43 - 1.07) $\mu m^3\ cm^{-3}$. The statistical significance test gave a p-value of virtually null, indicating that the differences are significant. In particular, the difference of about 26% in aerosol volume, combined with the similar number concentrations, indicates that the fine mode aerosol particles closest to the canopy are smaller than those at 325 m. These results are well-aligned with recent findings of the continuous appearance of sub-50 nm aerosol particles, observed as frequent nanoparticle bursts close to the canopy, which are likely associated with new particle formation processes.

The results for the aerosol volume agree with the medians of the aerosol size distributions at the two height levels, shown in Figure S6. The peak representing the Aitken mode at 60 and 325 m has the same geometric mean diameter of $D_g = 68.5$ nm. In contrast, the accumulation mode at the two heights showed differences that explain the volume difference. At 60 m the



peak of the accumulation mode is centered at $D_g = 143.7$ nm, while at 325 m the peak is centered at $D_g = 162.5$ nm. The larger mean diameter of the accumulation mode particles at 325 m is associated with aging processes and upward transport inside the PBL, which may result in differences in the coating of these particles. The gradient observed in the aerosol fine mode volume may also imply different scattering and absorption efficiencies in the vertical profile, likely related to different

atmospheric processes and meteorological transport. The scattering and absorption efficiencies will be explored in more detail in section 3.5.

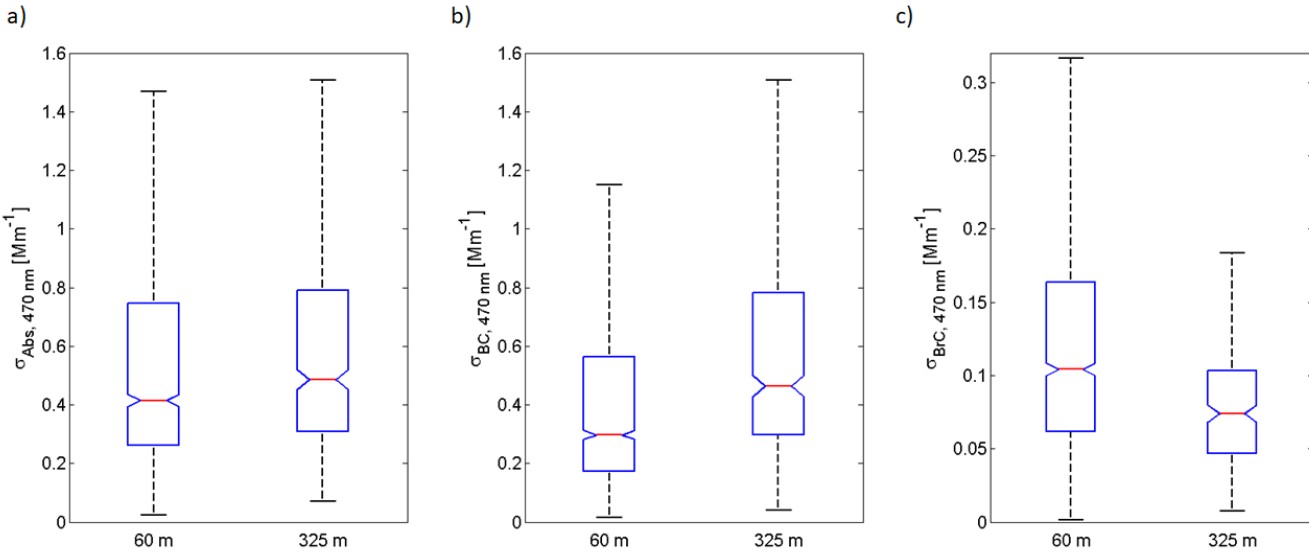

**Figure 4.** Comparative boxplots of a) absorption coefficients measured by the AE33 at 470 nm, b) absorption coefficients exclusively due to the black carbon (BC) component, and c) absorption coefficients exclusively due to the brown carbon (BrC) component at 60 and 325 m height. The box represents the quartiles, the whiskers represent the 90th and 10th percentiles, and the horizontal line represents the median.

An interesting aspect of the aerosol absorbing properties was observed by calculating the brown carbon (BrC) component using the AE33 data and a methodology similar to that described in Morais et al. (2022). This method derives the BrC component at 470 nm, using the AAE calculations with two different pairs of wavelengths and the expected BC value obtained with

Mie code simulations. Figure 4 presents the boxplots with comparative results for the total absorption coefficient, measured at 470 nm, and the individual absorption coefficients for black carbon (BC) and BrC at 470 nm at both height levels.

The results show different contributions of BC and BrC at the two heights. At the 60 m level, the median and IQR of the BrC absorption coefficient were 0.10 (0.06 - 0.16) Mm$^{-1}$, whereas at the 325 m level, the median and IQR were 0.07 (0.05 - 0.10) Mm$^{-1}$. In contrast, the coefficients for BC were higher at 325 m, with a median and IQR of 0.46 (0.30 - 0.78) Mm$^{-1}$,

while at 60 m, the median and IQR were 0.30 (0.17 - 0.56) Mm$^{-1}$. In percentage comparisons, the contribution of BrC at 60 m is approximately 27%, which agrees with previous studies (Saturno et al., 2018b), while at 325 m, this contribution is only 14%, which is a remarkable contrast. The results show that close to the canopy, the contribution of BrC to the total absorption



coefficient at 470 nm is approximately twice that at 325 m, while at the highest level, the contribution of BC is dominant. This may be related to the presence of fresher and likely newly formed aerosols at 60 m, which contain a higher fraction of BrC (Updyke et al., 2012). At 325 m, where the BC contribution is higher, aerosols are likely to be more influenced by the free troposphere, mainly during the night as the residual layer forms, and by long-range transport from Africa, which brings more

aged aerosols to the site, which are high in BC and where BrC has been lost due to photochemical aging (Holanda et al., 2020, 2023).

The differences in the coating of these aerosol populations between the two heights could indicate different processing stages. Particles with larger volumes also result in larger scattering and absorption cross-sections. This could partially explain the spectral differences obtained in the vertical profile, with more significant differences in the scattering coefficients and the

SAE for the aerosol populations at both height levels. The occurrence of low-level jets (LLJ) in the Amazon region (Anselmo et al., 2020) may be a source of aged aerosols. These LLJs could act by bringing to the surface aged aerosols from the aerosol atmospheric rivers above the forest, a process that, in the Amazon, occurs mainly during the night (Chakraborty et al., 2022). Another possibility is the presence of a more important coarse mode at 60 m, as highlighted by different studies (Prass et al., 2021; Barbosa et al., 2022).

### 3.3   Diurnal cycle

The median diurnal cycles of the optical coefficients and the aerosol volume are shown in Figure 5. During the daytime, when the boundary layer reaches its highest vertical extension and when turbulent processes are maximized, the scattering and absorption coefficients are identical at 60 and 325 m, with median scattering coefficients at 525 nm of about 8 $Mm^{-1}$. The absorption coefficients are about 0.25 $Mm^{-1}$ at 14 LT when they reach their highest values. The single-scattering albedo,

$\omega_{0,637nm}$, does not vary significantly throughout the day, with similar values for both heights of approximately 0.94.

Figure S7 shows the median diurnal cycles of particle number concentrations in each of the three aerosol size modes: sub-50 nm, Aitken, and accumulation mode. As in Figure 5, the diurnal behavior is quite evident. Following the results of Franco et al. (2022), the sub-50 nm particle fraction has an interesting behavior: particle number concentrations tend to increase between 14 LT and 8 LT, with medians of 60 and 35 $cm^{-3}$, respectively for 60 and 325 m. The concentrations decrease from 8 to 14 LT,

with minimum values of 35 and 15 $cm^{-3}$, respectively, to 60 and 325 m. These results agree with literature results (Machado et al., 2021). Differences in concentrations in the vertical profile, as shown in Figure S7a, are observed throughout the day. The possibility of an aerosol source close to the canopy has been recently discussed by Machado et al. (2023). This study shows the occurrence of frequent nanoparticle bursts close to the canopy and describes how the chemical species necessary for SOA formation are entangled, which likely indicates NPF mechanisms close to the canopy.

The diurnal cycles of the accumulation mode (Figure S7c) have the opposite trend to the sub-50 nm particles but the same trend as the aerosol volume, where concentrations decrease during the night until the beginning of the day and then start growing again with the evolution of the PBL as the photochemical processes provide conditions for particle growth. In particular, during the daytime, the accumulation mode particle number concentrations at both height levels are similar, showing the effectiveness of mixing in the PBL, in particular when the PBL reaches its greatest height, at approximately 13-14 LT, with a





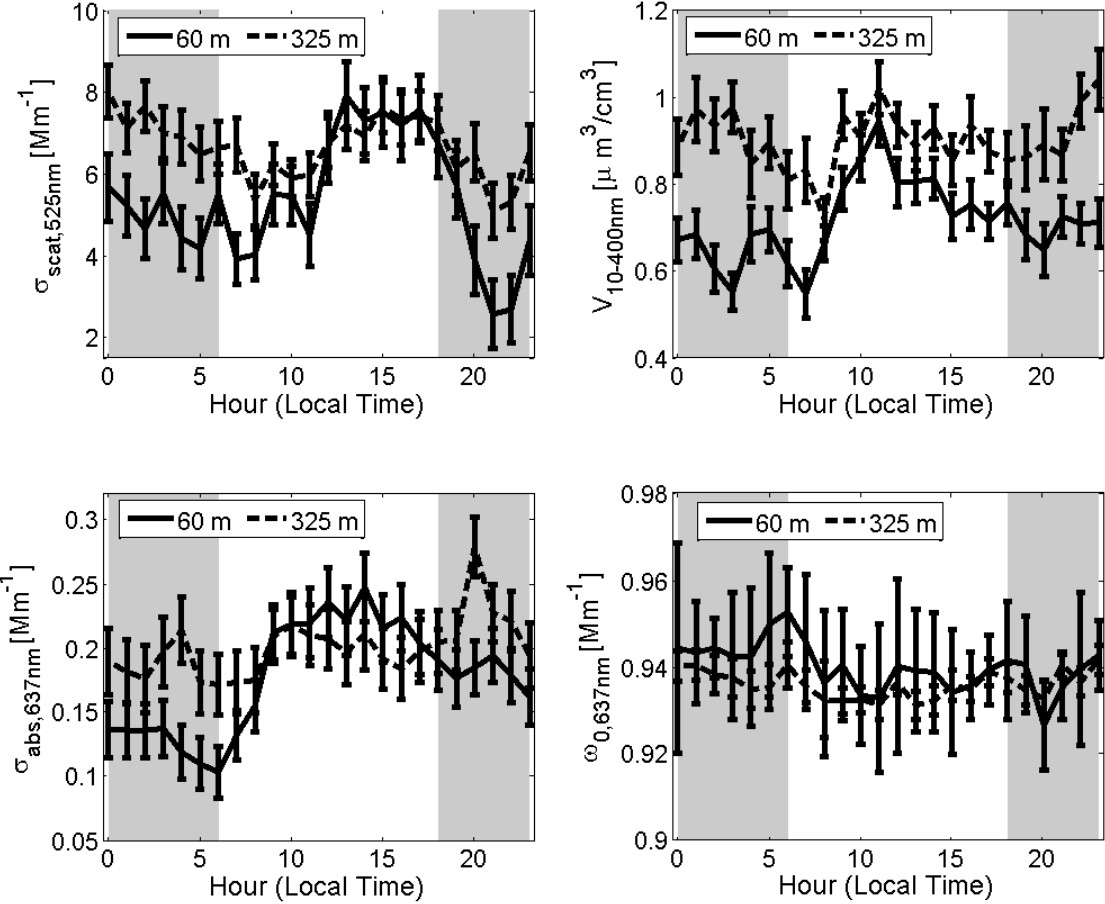

**Figure 5.** Median diurnal cycle of a) aerosol scattering coefficient, $\sigma_{scat,525nm}$, at 525 nm, b) particle number concentration ($N_{10-400nm}$), c) absorption coefficient, $\sigma_{abs,637nm}$, at 637 nm, and d) single-scattering albedo, $\omega_0$, calculated at 637 nm. Gray bars represent nighttime. Error bars are the standard error.

median concentration of about 200 cm$^{-3}$. Note that the aerosol optical coefficients also follow a similar trend (Figure 5c-d), showing that the optically active aerosols are those of accumulation mode.

In contrast, during the nighttime, there are clear differences in all aerosol physical properties. After establishing the stable nocturnal boundary layer, whose height is below 300 m (Fisch et al., 2004; Carneiro and Fisch, 2020), both absorption and scattering coefficients are typically higher at 325 m. The scattering coefficient shows differences up to $\sim 3$ Mm$^{-1}$ and the absorption coefficient presents differences of $\sim 0.1$ Mm$^{-1}$ between the levels. The aerosol volume also shows clear differences, with higher medians at 325 m, mainly during the nighttime. Also, there is a clear separation between the accumulation mode particles at the two height levels, which explains the differences in the optical properties and volume. The accumulation mode at 60 m decreases more strongly in the nocturnal BL, indicating dry deposition to the canopy during the night. Aitken mode





concentrations are quite stable at 60 m, with a median value of about 120 cm$^{-3}$. In comparison, at 325 m, there is a sudden decrease in concentration at around 8 LT, increasing again after sunset.

The observed differences in the diurnal cycles are likely related to mixing processes that can still occur at 325 m during the nighttime within the residual layer, which ages the aerosol, increasing its absorption and scattering cross-sections. In addition,

the processing of aerosols in clouds and fog is more likely to occur at the highest level, unlike at 60 m, since the stable layer does not contribute efficiently to the aging of aerosol particles. Another possibility, as presented in section 3.2 is the nocturnal occurrence of LLJ, which could bring aged particles to the residual layer. Therefore, the differences in the diurnal concentration profiles suggest that different processes can influence the aerosol population at each height level. More detailed investigations are suggested for future studies, seeking to identify and characterize in detail the sources and sinks of these particles and the

importance of meteorology in the vertical profile of the PBL in the central Amazon.

## 3.4   Downdraft events

Weather events modify the physical and chemical properties of aerosols through various processes. The main ones can be summarized as the wet deposition of particles during precipitation and in clouds, injection of aerosol populations with different characteristics into a specific layer of the atmosphere through downdraft and updraft mechanisms, and transport of air masses

through advection, as already discussed for aerosol number concentration by Machado et al. (2021). This section presents how intense downdraft events affect the aerosol population in the vertical profile of the lower troposphere at the ATTO site during the wet season, focusing on their optical properties.

Using the downdraft detection algorithm presented in Section 2.8, 120 strong convective events were identified. Furthermore, a composite analysis was performed on the data set, similarly to Machado et al. (2021), regarding the relative position of the

data points to the onset of the closest detected downdraft event. Thus, it was possible to explore the average effect of intense convective events on the aerosol properties. Figure S13 shows the average behavior of the potential temperature anomaly and precipitation before and during intense convective events. It can be seen that, on average, the potential temperature drops rapidly for 1.5 hours and then slowly rises at a rate of $(0.30 \pm 0.03)$ K h$^{-1}$. Furthermore, precipitation increases at the event's onset and reaches its maximum half an hour later, decreasing rapidly but remaining above the reference (before the event) for

up to 5 hours.

Figure 6 shows how convective events influence the aerosol populations observed at 60 and 325 m, reducing particle absorption and scattering through wet deposition of larger particles when there is associated precipitation. As previously shown, the aerosol load is higher at 325 m, reflected in a higher absorption coefficient for $\Delta t < 0$ in Figure 6a. However, when a convective event begins, the associated precipitation removes a large portion of the absorbing particles from the atmosphere at

both heights, equalizing the values for a short period. Quickly after the decrease in precipitation, the absorption at both heights differs again, with a more pronounced growth in the values measured at 325 m, of $0.026 \, Mm^{-1}h^{-1}$, against $0.011 \, Mm^{-1}h^{-1}$ at 60 m above ground level. The scattering coefficient, however, remained very low during the entire period, revealing that very clean conditions prevailed. In these conditions, the average effect of downdraft events was less impactful due to the lower amount of optically active particles in the atmosphere. There is still a noticeable decrease in the scattering coefficient before





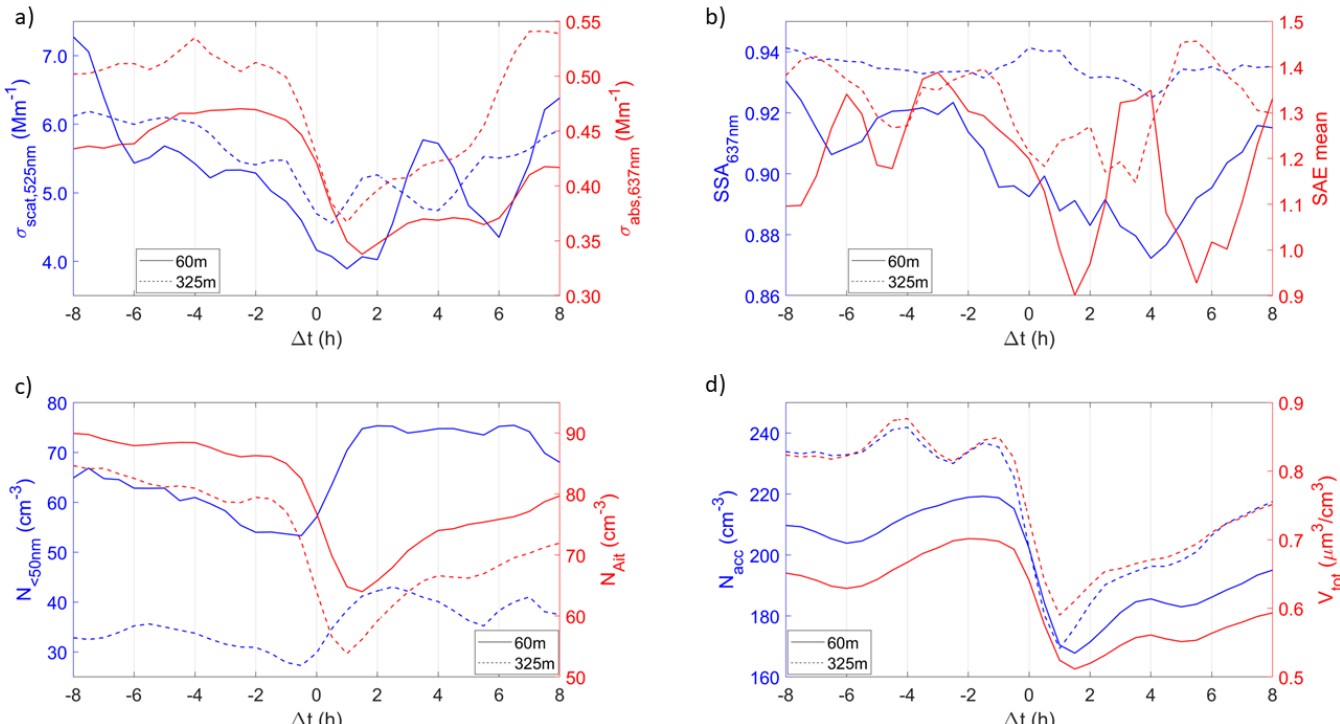

**Figure 6.** Composite analysis, considering the relative position of the data points to the onset of the closest detected downdraft event, for the aerosol properties in the vertical profile at ATTO using an event-normalized time stamp. (a) 525 nm scattering and 637 nm absorption coefficients, (b) single scattering albedo at 637 nm and mean scattering Ångström exponent, (c) Sub-50nm and Aitken mode concentrations, and (d) accumulation mode concentration and fine mode aerosol volume. The solid lines represent the measurements at 60 m and the dotted lines at 325 m.

the downdrafts and a slow climb after the events' onset. However, more measurements are required to increase the number of events in this analysis.

The study of the intensive properties of the aerosols yielded results that indicate a more profound change in the aerosol population after downdrafts and that the observed trends are not entirely explained by the reduction in number concentrations. The composite analysis indicated that SSA values at 325 m remain relatively stable at around 0.94 and are less affected by downdrafts than the SSA at 60 m, which decreases from 0.92 before the event onset to 0.88 a few hours after. Thus, despite the increase in absorption and scattering coefficients as the aerosol population recovers from the weather events, the scattering coefficient rises more slowly, which leads to a lower SSA value than before the convective event, as shown in Figure 6b. This conclusion is consistent with the chemical composition results, where the black carbon fraction during convection conditions is slightly higher than in stable atmospheric conditions. The SAE composite analysis is inconclusive due to the period's low amount of reliable nephelometer data. However, it is noticeable that SAE fluctuated less before the downdrafts and decreased





around $\Delta t = 0$ at both heights, with less variation at 325 m, reaching previous levels in up to 4 hours, unlike the other properties, which take longer on average to return to the pattern found under stable atmospheric conditions. In addition, the decrease observed in the Ångström scattering coefficient as soon as a convective event begins indicates a predominance of secondary organic aerosols and larger particles primarily emitted by the vegetation.

The relationships between precipitation and convection with the PNSDs were also studied using composite analyses for simultaneous measurements taken at both heights, with an approach similar to Machado et al. (2021). During downdraft events, the injection of small ultrafine particles from upper layers of the atmosphere and the increased ozone concentration that also favors the oxidation of VOCs is responsible for an increase of approximately 50% in the concentration of sub-50nm mode particles at both measurement heights. The number concentration values increased from 28 to 43 $cm^{-3}$ at 325 m and

from 53 to 78 $cm^{-3}$ at 60 m in the 2 hours following the onset of the downdraft events (Figure 6c). In contrast, the other two modes experience decreases in their average concentrations after convective events since precipitation can remove particles in these size ranges through wet deposition scavenging. After the downdrafts, the Aitken mode shows a strong increase of 2.5 $cm^{-3}h^{-1}$ at both heights.

On the other hand, the number concentration in the accumulation mode shows different post-downdraft growth rates at

60 and 325 m, which are 3.8 and 6.2 $cm^{-3}h^{-1}$, respectively. We also observed an increase in particle concentration in the accumulation mode at 60 m before the events, contrasting with a slight decrease in the Aitken mode, indicating that particle growth processes occur under stable atmospheric conditions. This is also reflected in the increase of the fine mode volume without an increase in total number concentration (Figure 6d). After the events, the total aerosol volume increases because the high concentration of sub-50 nm particles increases in diameter through oxidation and hygroscopic growth. However,

the total number concentration also increases at both heights as the aerosol population recovers from the events, suggesting the occurrence of biogenic emissions. Using linear regression, we can estimate particle number increase rates for the total concentrations at both heights of around 5.7 $cm^{-3}h^{-1}$ at 60 m and 7.9 $cm^{-3}h^{-1}$ at 325 m.

### 3.5 Vertical gradient of mass scattering and absorption efficiencies

This section explores the mass scattering and absorption efficiencies ($\alpha_{scat}(\lambda)$ and $\alpha_{abs}(\lambda)$, respectively) of the aerosol pop-

ulations in the vertical profile. Understanding these efficiencies is crucial for assessing how aerosols influence the balance of incoming and outgoing radiation in the Earth's atmosphere. The scattering and absorption efficiencies are calculated through the slope of the fitting curve between the scattering/absorption coefficient at a given wavelength and the mass concentration. Approximate mass concentrations used in the analyses were retrieved from the aerosol volume at each height (within the size range of 10 to 400 nm) multiplied by the average total loading of the wet season, of 1.4 $\mu g\ m^{-3}$, which was obtained by pre-

vious studies at the ATTO site (Artaxo et al., 2022). This procedure was necessary since, in the analyzed periods, no chemical concentration measurements were obtained simultaneously at the two heights with similar instruments. A similar approach was also used by Ponczek et al. (2022). Figure S8 shows the variability of the scattering coefficients at 525 nm, $\sigma_{525m,}$, as a function of the mass concentration, where the colors in panels a) and b) represent $\omega_{0,637nm}$, and in panels c) and d) represent total concentrations, $N_{10-400nm}$. As in the previous section, comparative analyses were obtained for periods when the instru-



ments measured simultaneously at 60 and 325 m. In Figure S8, there is more variability of the scattering efficiency at 60 m, as the slopes of the curves that could be fitted to adjust the data change more frequently according to different atmospheric conditions related to different aerosol populations. We also find that $\omega_{0,637nm}$ and $N_{10-400nm}$ shows more variability during specific aerosol events at 60 m, whereas they are more stable at 325 m.

The period from 8 to 19 May 2019 was chosen as representative clean days to assess the differences in scattering efficiencies in the vertical profile, now considering all wavelengths measured by the nephelometer (450, 525, and 635 nm). In addition, this period presents clear linearity in $\sigma_{scat} \times$ mass concentration curves that was not affected by strong episodic events, as shown in Figure S10. The panels a) to c) show the measurements at 60 m height, and the panels d) to f) show the measurements at 325 m height. Figure S10 corroborates that there is more dispersion in the scattering coefficient and mass concentration at 60 m, while
at 325 m, the linear relationship is more homogeneous. The result supports the hypothesis that the 325 m aerosol population is more processed and aged than near the canopy. The spectral dependence is also clear in Figure S10. Shorter wavelengths have higher scattering efficiencies: at 60 m, we obtained $\alpha_{scat}(\lambda) \pm SE$ = 4.15±0.11, 5.49±0.11 and 7.74±0.12 $m^2\ g^{-1}$ for 635, 525, and 450 nm, respectively. At 325 m, the observed variability was $\alpha_{scat}(\lambda) \pm SE$ = 2.46±0.04, 3.76±0.05 and 5.26±0.06 $m^2\ g^{-1}$ for 635, 525, and 450 nm, respectively.

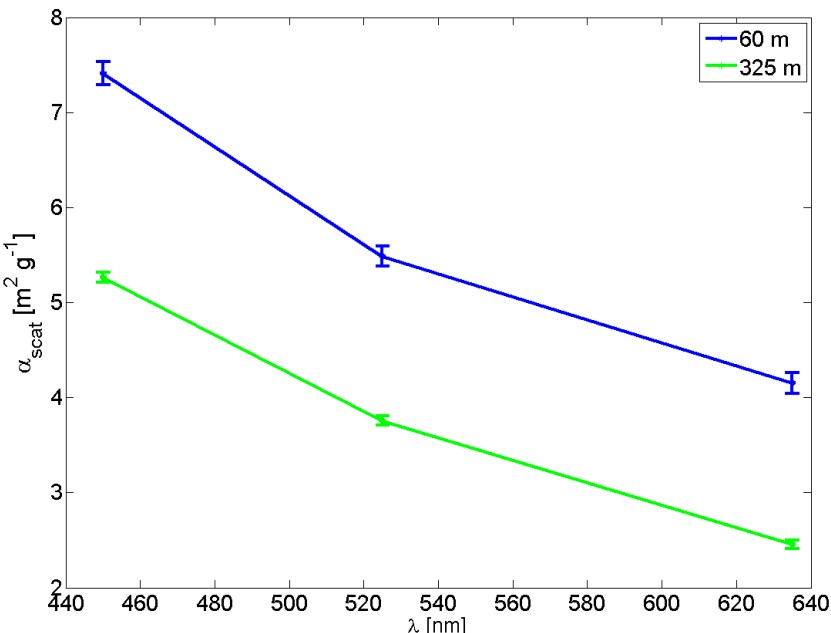

**Figure 7.** Mass scattering efficiency, $\alpha_{scat}(\lambda)$, as a function of wavelength for measurements at 60 and 325 m height.

A direct comparison between the scattering efficiencies at 60 and 325 m as a function of wavelength is presented in Figure 7. The difference in scattering efficiency with height is evident. Aerosol populations near the canopy are more efficient at



scattering radiation than at 325 m for all wavelengths. The percentage differences vary with wavelength, but on average, they are approximately 35%. The results suggest that aerosols at 325 m, as observed in Figure S6, are likely to be more processed and are less efficient in scattering radiation. In contrast, smaller aerosols in direct contact with fresh VOC emissions from vegetation are more likely to scatter radiation more efficiently. Another possibility that might explain the results is that the

higher apparent MSE at 60 m is likely due to the presence of a coarse mode, which is not detected by the SMPS. In fact, Prass et al. (2021) observed that bioaerosols account for about 70% of the aerosol coarse mode at ATTO, with higher concentrations at 60 m, which decreases with height. Figure S9 presents the absorption coefficient, $\sigma_{abs,637nm}$, as a function of the mass concentration in a representation similar to Figure S8. The figure shows a peculiar behavior of the coefficient variability at 60 and 325 m height. In contrast to the scattering efficiencies case, the absorption coefficient varies significantly less at 60 m. As

obtained from the comparisons in Figure 4, this is likely due to the influence of long-range aerosol transport at 325 m, which increases the BC concentration more than at 60 m.

Figures S9a and b suggest that the differences in the MAE values, as reflected in the slopes of the fitting curves, are related to differences in SSA, whereas aerosol concentration does appear to have a significant influence, as shown in Figures S9c and d. The median $\omega_{0,637nm}$ at 60 m is 0.96, compared to the median at 325 m, of 0.92. Figure S9b, in particular, shows

a pronounced increase of MAE with decreasing SSA. The results suggest the influence of different aerosol sources on the physical characteristics in the vertical profile: the aerosol population closer to the canopy is, for instance, less susceptible to long-range transport events than the population at 325 m. Also, the differences regarding day and nighttimes discussed in section 3.3 could play an important role. The difference between this result and the one obtained in the comparisons in Figure 3 is that here we consider exclusively the periods in which the absorption and scattering coefficients and the aerosol

size distributions were measured simultaneously in the same time interval, at 60 and 325 m. In the case of Figure 3, as already mentioned, comparisons were performed for periods in which similar instruments that measure the same physical property operated simultaneously.

A comparison of the MAEs, $\alpha_{abs,637nm}$, in the vertical profile was made for the same period considered for the mass scattering efficiencies: 8 to 19 May 2019. The absorption coefficients at 637 nm measured by the MAAPs were considered for

the $\alpha_{abs,637nm}$ calculations. Figure S12 presents the absorption coefficient at 637 nm as a function of mass concentration, with linear fits for 60 and 325 m. In contrast to the scattering efficiencies, which showed pronounced differences between the two heights, the absorption efficiencies were very similar, with $\alpha_{abs,637nm}$ of 0.210±0.003 and 0.226±0.004 $m^2\ g^{-1}$ at 60 and 325 m , respectively. As with the spectral dependence of the absorption coefficients shown in Figure S5, $\alpha_{abs,637nm}$ does not vary significantly with height. This indicates that, although the results in Fig. 4 show more influence of the BC component at

325 m and BrC at 60 m, the mass absorption efficiency of the total absorption coefficient is similar for the aerosol populations at both heights.

However, contrasting results are obtained when analyzing the BC and BrC components of the total absorption coefficient separately. Figure S14 shows comparative analyses obtained with the BC and BrC components at 60 and 325 m height from 8 to 18 May 2020. This period was chosen because AE33 measurements are available at both heights, and it represents clean

atmospheric conditions, with no clear particle events that could change the linearity of $\sigma_{abs,470nm}$ vs. the mass concentration.



Panels a) and b) in Figure S14 show the absorption coefficients exclusively due to BrC as a function of mass concentration for 60 and 325 m height, respectively. At 60 m (panel a), there is no apparent correlation between the variables, so it was not possible to estimate $\alpha_{BrC,470nm}$ in this case. The lack of correlation at 60 m is likely related to the high variability of the aerosol population, which is influenced by fresh primary and secondary biogenic aerosols.

At 325 m (panel b), despite some dispersion, it was possible to obtain a linear fit and estimate the mass absorption efficiency due exclusively to BrC, $\alpha_{BrC,470nm}$ = 0.049±0.007 $m^2\ g^{-1}$, despite the relatively low correlation, with $R^2$ of 0.4. The BrC absorption coefficients are significantly smaller than the total absorption coefficients and very close to the detection limit of the technique. This increases the coefficient estimation uncertainty, which may explain the low correlation. The low absorption efficiency of BrC at 325 m compared to the total mass absorption efficiency indicates that the BrC component contributes little
to the total absorption and has a much less significant radiative effect than BC.

Panels c) and d) in Figure S14 show the absorption coefficients exclusively due to BC as a function of mass concentration for 60 and 325 m height, respectively. Linear fits were obtained for both cases, with satisfactory correlation coefficients of $R^2 > 0.65$. The results show clear differences in BC's mass absorption efficiencies in the vertical profile. At 60 and 325 m, $\alpha_{BC,470nm}$ = 0.28±0.02 and 0.55±0.04 $m^2\ g^{-1}$, respectively. The almost 50% difference in efficiencies highlights BC's strong
absorption efficiency in the aerosol population at 325 m height, even during the cleaner months of the wet season. The result reinforces the hypothesis that long-range transport strongly influences this population, bringing dust and biomass burning from West Africa and Northeast Brazil to the central Amazon (Talbot et al., 1990; Wang et al., 2016). The BC influence is less significant close to the canopy than at 325 m, most likely because the population of biogenic aerosols is more prevalent there.

### 3.6   Effective aerosol refractive index

Figure 8 presents the results for the effective aerosol refractive index at 637 nm obtained at 60 (blue) and 325 m (red) heights. The statistical analysis showed a strong correlation between the measured and modeled optical coefficients, with $R^2$ values of 0.99 and 0.98 for scattering and absorption coefficients, respectively, where all p-values were statistically significant (p-values < 0.01). The results indicate that the scattering component of the refractive index, **n**, is, on average, higher at 60 m than at 325 m, with values of 1.33 and 1.27, respectively. The diurnal cycle of **n** also reflects this difference, with higher values at 60 m,
particularly in the early morning, as the differences are less pronounced during the afternoon with the PBL mixing the air very effectively.

These results also agree with the higher radiation scattering efficiency at 60 m, reinforcing that the aerosol populations at 60 and 325 m heights have different radiative impacts. In contrast, the real component of the refractive index remains relatively constant at 325 m throughout the day, suggesting that the aerosols at this height suffer less from fresh emissions and are likely more aged than those at 60 m. Higher values of **n** at 60 m reinforce the higher mass scattering efficiency observed at this height.
The results show that the aerosol particles at 60 m are more efficient in scatter radiation than those at 325 m. Moreover, the imaginary component of the refractive index, **k**, of natural aerosols is similar for both heights, with a mean value of 0.006 and nearly constant throughout the day. This particular aspect also explains the similar absorption coefficient at both heights.



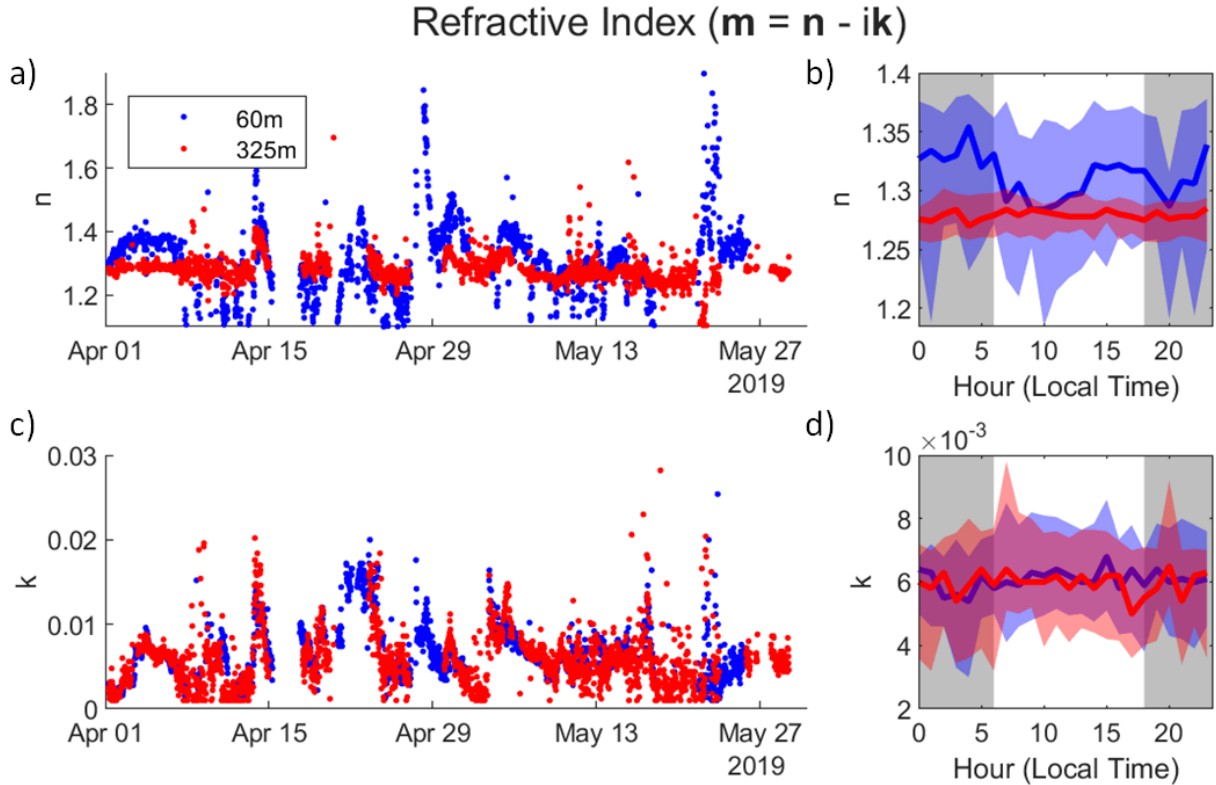

**Figure 8.** Equivalent aerosol refractive index **m** at 637 nm. Panel a) shows the time series for the real part (n), while panel c) shows the same for the imaginary part (k). Panel b) shows the median diurnal cycle for the real component and d) for the imaginary part. The solid line indicates the median and the semi-transparent area indicates the interquartile range. Results at 60 m height are shown in blue, while those at 325 m are in red.

Guyon et al. (2003) retrieved the effective refractive indices for aerosols near the surface during the wet season in a remote Amazonian forest site. The study obtained **m** = 1.42(0.04) - 0.006(0.003), which is close to the values observed for 60 m height. The slightly higher value of **n** in Guyon et al. (2003) is likely associated with primary emissions directly from the ground, whereas at 60 m height, the primary canopy emissions are likely playing a significant role on the effective **n**. The imaginary part of the refractive index in both studies was identical. Similarly to our study, Guyon et al. (2003) also observed a pronounced diurnal variability of **n** likely associated with fresh emissions from the vegetation, with the imaginary part showing much less variability, which is in accordance with our result. It is worth mentioning that both Rizzo et al. (2013) and Guyon et al. (2003) observed a much higher **n** during the dry season, which is associated with fresh biomass-burning emissions. Differently from what is observed, for instance, at 325 m, the lower **n** at this level is likely due to the aging processes and not due to BC. The results emphasize the different radiative and climatic effects of biomass burning and natural aerosols, which can be addressed in more detail in future studies.



## 4  Conclusions

This study provides new insights into the variability of physical aerosol properties in the vertical profile of the lower troposphere in the central Amazon. The extensive aerosol properties, such as aerosol volumes and scattering coefficients, showed significant variability in the vertical profile. The analysis of diurnal cycles revealed differences in particle number concentrations of each size mode, indicating a strong efficiency of the boundary layer in mixing aerosols during the day and the divergent evolution of aerosol populations during the night, separated by the establishment of the stable nocturnal boundary layer. We also found a pronounced difference in the magnitude and spectral dependence of the scattering coefficient in the vertical profile. In contrast, no such difference was detected in the absorption coefficients measured at different wavelengths.

We found and characterized the changes in the aerosol populations before and after downdraft events at the two measurement heights. The absorption and scattering coefficients decrease as optically active particles are removed by wet deposition. Absorption quickly increases again after the events, but the recovery is faster at 325 m than at 60 m, while scattering remains lower for several hours. Similarly, the concentrations of the Aitken and accumulation modes decrease and recover quickly after the end of precipitation, with similar particle number growth rates between the two heights in the Aitken mode and slightly faster recovery at 325 m in the case of the accumulation mode. In addition, convective events were concomitant with an increase in sub-50 nm mode particles, whose concentration rapidly increased by up to 50% at both heights.

The total volume of aerosol particles, generally larger at 325 m, is practically equalized during convective events, but the post-downdraft increase differs between the two heights, separating the curves. These results show that vegetation highly influences the atmosphere close to the canopy during clean conditions, and forest emissions largely determine the response to weather events. At the same time, at 325 m, the physical properties of the aerosols are less determined by how the ecosystem responds; thus, the return to usual conditions is faster. Intensive properties also change, with a decrease in SSA values after the events, as absorption increases quickly and scattering remains lower. Finally, the scattering Ångström coefficient quickly drops as the events begin but then rises again, reaching previous levels within 4 hours, unlike the other properties, which on average take longer to return to the values found under stable atmospheric conditions, although future studies would benefit from further investigations of this aspect.

The estimation of mass scattering and absorption efficiencies and their variability in the vertical profile of the troposphere in the central Amazon is an unprecedented and climatic-relevant component of this study. The results showed that the aerosol population close to the canopy had a higher mass scattering efficiency than at 325 m, both with a clear spectral dependence. This difference could be explained by the different size distributions at each height, which shows more small ultrafine particles at 60 m. These fresh particles are transported to the ATTO upper levels and age along the path, increasing in size until reaching the measurement height of 325 m. Considering the total absorption coefficient, the mass absorption efficiency was similar at 60 and 325 m, approximately 0.21 $m^2\ g^{-1}$.

The equivalent aerosol refractive index shows a higher scattering component at 60 m than at 325 m, reflected in the refractive index's diurnal cycle. The real component remains constant at 325 m and increases during the night at 60 m, suggesting more aged aerosols are constantly found at the higher levels. Furthermore, the imaginary component is similar for both heights with



no significant diurnal trend. These results corroborate the mass scattering and absorption efficiencies obtained on the vertical profile. The findings of an aerosol population with a lower refractive index on the real component at higher levels, while the particles closer to the canopy scatter radiation more efficiently, is fundamental in global climate modeling to represent better the aerosol component of the Earth system on tropical regions. In conclusion, the study provides new insights into the variability

of aerosol properties in the vertical profile of the troposphere in the central Amazon, which is essential for understanding the impact of aerosols on climate. The findings of this study can be used to improve aerosol models and their representation in climate models, ultimately contributing to more accurate predictions of future climate change.

*Data availability.* The data of the key results presented here have been deposited in supplementary data files for use in follow-up studies. Particle number size distributions for the entire observation period are available via the ATTO data portal at https://www.attoproject.org/.

Please refer to the corresponding authors for requests beyond the available data.

*Author contributions.* MAF and RV contributed equally to this study. MAF, PA, and CP designed the study. MAF, RV, BH, BBM, FM, LK, and DW processed and analyzed the ATTO data. LATM, MOA, LVR, JPN, MAC, UP, and SC contributed with valuable ideas and comments to the analysis and the manuscript. MAF and RV wrote the manuscript. All authors contributed to the discussion of the results as well as the finalization of the manuscript. PA and CP supervised the study.

*Competing interests.* At least one of the (co-)authors is a member of the editorial board of Atmospheric Chemistry and Physics.

*Acknowledgements.* Marco A. Franco acknowledges the CNPq scholarship (project 169842/2017-7), the CAPES sandwich doctorate program (project 88887.368025/2019-00) and FAPESP fellowship (21/13610-8). Rafael Valiati acknowledges the CAPES scholarship (88887. 611308/ 2021-00). Bruna A. Holanda acknowledges the CNPq (process 200723/2015-4) and the Max Planck Graduate Center with the Johannes Gutenberg University Mainz (MPGC). The ATTO research has been supported by the Max Planck Society, the German Federal

Ministry of Education and Research (BMBF contracts 01LB1001A, 01LK1602A, 01LK1602B, and 01LK2101B), the Brazilian Ministério da Ciência, Tecnologia e Inovação (MCTI/FINEP contract 01.11.01248.00), the FAPESP (Fundação de Amparo à Pesquisa do Estado de São Paulo) (grant no. 2017/17047-0, 2022/07974-0), the CNPq project (grant no. 169842/2017-7, 88887.611308/2021-00), the CAPES project (grant no. 88887.368025/2019-00), LBA/INPA, the Amazon State University (UEA), FAPEAM, and SDS/CEUC/RDS-Uatumã. We acknowledge the support by the Instituto Nacional de Pesquisas da Amazônia (INPA). We would like to thank the ATTO team members,

including Reiner Ditz, Jürgen Kesselmeier, Susan Trumbore, Alberto Quesada, Thomas Disper, Thomas Klimach, Andrew Crozier, Björn Nillius, Uwe Schulz, Steffen Schmidt, Delano Campos, Sam Jones, Fábio Jorge, Stefan Wolff, Juarez Viegas, Sipko Bulthuis, Francisco Alcinei Gomes da Silva, Valmir Ferreira de Lima, Feliciano de Souza Coelho, André Luiz Matos, Davirley Gomes Silva, Torsten Helmer, Karl Kübler, Olaf Kolle, Martin Hertel, Kerstin Hippler, Hermes Braga Xavier, Nagib Alberto de Castro Souza, Adir Vasconcelos Brandão,





Amauri Rodriguês Perreira, Antonio Huxley Melo Nascimento, Roberta Pereira de Souza, Bruno Takeshi, Wallace Rabelo Costa and all further colleagues involved in the technical, logistical, and scientific support within the ATTO project.



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
