# Peer review of "Vertically resolved aerosol variability at the Amazon Tall Tower Observatory under wet season conditions"

_EGUsphere, 2023_

## Author Comment (AC1)

**Responses to Reviewer #1 of the manuscript "Vertically resolved aerosol variability at the Amazon Tall Tower Observatory under wet season conditions" by Franco & Valiati et al., submitted for publication in Atmospheric Chemistry and Physics**

Dear Editor, we would like to thank reviewer #1 for the valuable comments and useful suggestions to improve our manuscript. Below, you can find answers and actions for each individual comment. To make it easier to identify the individual answers and actions, we used the following color code strategy:

- **In black are the reviewer's comments.**
- **In blue are the author's responses.**
- ***In blue and italics are the text modifications we made in the manuscript.***

General comments:

This article provides and analyzes very valuable information of the sub-micron aerosol in a remote Amazon site. It informs not only number concentration and size distribution (10 nm to 400 nm) but also extensive and intensive optical properties of the aerosols, at two different heights, thanks to the well known ATTO tower. This kind of data is very scarce in the region, since most of the (few) air quality monitoring stations in South America are placed in urban regions.

As the authors claim, comparing continuum measurements at different heights is very important, since it provides information regarding sources (new particle formation, biogenic emissions, long range transport) and helps validate and improve aerosol models and remote sensing retrievals. The location of the site allows to study a pristine Amazonian atmosphere, with impact of Africa long range biomass burning, and in addition provides insight regarding the effect of downdraft events.

For instance, they observe the higher relevance of BC at 325 m due to the effect of the African plume (and optical aging of BrC); they show new particle formation near the canopy (sub-50 nm) and growth during the day through aging (accumulation mode); they observe the rate of new particle formation after wet deposition; etc.

The article provides very relevant figures (both main article and supplement) that are adequate to derive the conclusions in the main text. The article is well written and supported with previous results from literature, adequately discussed.

I recommend publication after minor revision.

We thank Referee #1 for the very constructive comments and useful suggestions. They helped us to clarify important aspects of our discussions and, thus, to improve the manuscript overall.

Specific comments:

Section 2.3: if aerosol volume is estimated from SMPSs (as I imagine), it would be relevant to introduce in this section a comment on that, and maybe the equations that connect the measured size distributions with the aerosol volume (here or in the Supplement).

*Many thanks for the suggestion. Indeed, we used the SMPS data to retrieve the aerosol volume. Section 2.2 was updated, and we added the following text to detail the process:*

*The fine mode number concentration and the aerosol volume were obtained by integrating the PNSD. For the particle volume, it was considered that all particles are spherical, following single-particle characterizations developed with Amazonian aerosols (Wu Li et al., 2019).*

Section 2.4: I believe it would be important to make clear which cut size (if any) have each optical instrument. For instance, usually the aethalometers use PM2.5 size cut, but not always. From the discussion it seems that nephelometer and MAAP let coarse aerosols in (effect of large biogenic aerosols is discussed), but it is not clear what sizes are allowed.

*Thanks for pointing that out. All the optical instruments measured with PM2.5 size cut. We added the following line to the manuscript to highlight it:*

*All optical instruments operated with a size cut of 2.5 $\mu$m.*

Section 2.5: The adjustment of m and k with Mie's calculations is interesting. Do you think is reasonable to assume sphericity for these particle sizes? Do you have microscopy information for similar samples?

*Yes, sphericity is a reasonable assumption for submicrometer aerosols during the wet season at the ATTO forest site. Single-particle characterization of aerosols using ED-EPMA (energy-dispersive electron probe X-ray microanalysis) has shown that SOA (secondary organic aerosols) and ammonium sulfate comprised 73 %–100 % of submicrometer aerosols (Wu et al., 2019). The authors report that most submicron SOAs were internally mixed with ammonium sulfate, having a circular shape in the images. The overall circular morphology of SOA-ammonium sulfate mixed particles indicate that they are mostly in aqueous droplets at the time of sample collection rather than in crystalline form. Therefore, the spherical assumption is reasonable.*

Section 3.2: "Although differences were observed for the SAE and the vertical profile, indicating different aerosol coarse mode populations, the $\omega_{0,637nm}$ indicates

that the fine mode aerosol population in both highs is very efficient in scattering radiation." I would like the authors to comment a little further on this phrase (and other connected phrases below in the article). In the one side, I understand that you did not measure the coarse mode aerosols in this work, but that is plausible that there are more coarse biogenic particles at 60 m, which should be expressed as a lower a SAE. But here you see small differences in SAE in the vertical profile, and you assign it to the fact that "the fine mode aerosol population in both highs is very efficient in scattering radiation". Am I correct? Later in this section, you also suggest that the differences in the accumulation mode median diameter could influence the aerosol optical properties. Please comment a little bit more on your view regarding the impact of coarse and fine mode aerosols in scattering coefficient and SAE. Maybe comparing the effects that increase or decrease the scattering coefficient and SAE and the suggested balance among these factors/causes at the end of this section? It can also be connected with complementary results and discussion in Sections 3.5 and 3.6.

The first highlighted sentence was divided and more clearly phrased. The scattering efficiency and albedo are not necessarily related to the observed difference in the SAE at both heights.

Furthermore, the balance between the fine and coarse modes between the two heights has contrasting effects on the scattering coefficient and the SAE. For the scattering coefficient, the fine mode is likely more important than the coarse mode, due to the larger fraction of aerosol surface area being linked to fine mode particles. In this scenario, the larger fine mode volume observed at 325 m suggests that the particle cross-section is larger as well, which explains the higher scattering coefficient. On the other hand, the more prevalent coarse mode at 60 m affects the SAE greatly. Schuster et al. (2006) found that the effective diameter of the fine mode particles is less important than the fine/coarse mode ratio for the SAE, which is likely the reason that this intensive property is so different between the two height levels.

In order to improve this discussion, we included the following text at the end of Section 3.2:

*As most of the particles found above the canopy are within the fine mode range, the scattering coefficient is largely dependent on the surface area of fine mode aerosols, and less on the coarse mode mass concentration. This is likely the reason for the higher scattering coefficient at 325 m, as the larger particles have higher scattering cross-sections. Nevertheless, Schuster et al. (2006) found that the SAE is less dependent on the effective diameter of the fine mode particles, and more influenced by the balance between fine and coarse mode aerosols. A strong vertical gradient of coarse mode particles, more prevalent at 60~m, is likely the reason behind the large difference in the SAE.*

Section 3.5, p.18: "The results suggest that aerosols at 325 m, as observed in Figure S6, are likely to be more processed and are less efficient in scattering

radiation. In contrast, smaller aerosols in direct contact with fresh VOC emissions from vegetation are more likely to scatter radiation more efficiently. Another possibility that might explain the results is that the higher apparent MSE at 60 m is likely due to the presence of a coarse mode, which is not detected by the SMPS. In fact, Prass et al. (2021) observed that bioaerosols account for about 70% of the aerosol coarse mode at ATTO, with higher concentrations at 60 m, which decreases with height.". Are there previous results that suggest that "smaller aerosols in direct contact with fresh VOC emissions from vegetation are more likely to scatter radiation more efficiently"? Or do you believe that the second hypothesis is more likely?

In fact, both hypotheses are likely, and further analyses are required to discriminate which one is the more important. From the perspective of the fresher and smaller particles, they are mainly constituted by secondary organic aerosols (SOA) derived from the oxidized volatile organic compounds (VOCs). Holanda et al., 2023, have shown that, at 325 m high at the ATTO site, pristine aerosols, which are mainly formed by oxidized SOA (80% of the mass fraction), have a significantly higher single scattering albedo (SSA) than long-distance aged aerosols, which are larger, with a relevant coating size and are more absorbing, even during the wet season. The SSA, just like the mass scattering efficiency (MSE), is an intensive aerosol property, and both are directly correlated; the bigger the SSA, the bigger the MSE. The lensing effect may prevail for the coarse mode particles, turning the biological particles, which are the main aerosols in this particular size range, the main source of radiation scattering.

To improve the quality of this discussion, we added the following text to the manuscript:

*Holanda et al., 2023, have shown that, at 325 m high at the ATTO site, pristine aerosols, which are mainly formed by oxidized SOA (80% of the mass fraction), have a significantly higher single scattering albedo (SSA) than long-distance aged aerosols, which are larger, with a relevant coating size and are more absorbing, even during the wet season. The SSA, just like the mass scattering efficiency (MSE), is an intensive aerosol property, and both are directly correlated; the bigger the SSA, the bigger the MSE.*

*Another possibility that might explain the results is that the higher apparent MSE at 60 m is likely due to the presence of a coarse mode, which is not detected by the SMPS. In fact, Prass et al., 2021 observed that bioaerosols account for about 70% of the aerosol coarse mode at ATTO, with higher concentrations at 60 m, which decreases with height. Both hypotheses are likely, and further analyses are required to determine which is more relevant.*

Technical corrections:

P.18, "Shorter wavelengths have higher scattering efficiencies:"

It seems to me that the correct expression would be "Scattering efficiencies are higher for shorter wavelenghts"

(to my mind, aerosols HAVE higher or lower scattering efficiencies, not wavelenghts).

Thanks. We corrected the sentences in the manuscript.

**References**:

Schuster, G. L., Dubovik, O., and Holben, B. N.: Angstrom exponent and bimodal aerosol size distributions, Journal of Geophysical Research: Atmospheres, 11.

Wu, Li, et al. "Single-particle characterization of aerosols collected at a remote site in the Amazonian rainforest and an urban site in Manaus, Brazil." Atmospheric Chemistry and Physics 19.2 (2019): 1221-1240.

Rizzo, L. V., Artaxo, P., Müller, T., Wiedensohler, A., Paixão, M., Cirino, G. G., Arana, A., Swietlicki, E., Roldin, P., Fors, E. O., Wiedemann, K. T., Leal, L. S. M., and Kulmala, M.: Long term measurements of aerosol optical properties at a primary forest site in Amazonia, Atmos. Chem. Phys., 13, 2391–2413, https://doi.org/10.5194/acp-13-2391-2013, 2013.

Holanda, B.A., Franco, M.A., Walter, D. *et al.* African biomass burning affects aerosol cycling over the Amazon. *Commun Earth Environ* 4, 154 (2023). https://doi.org/10.1038/s43247-023-00795-5

---

## Author Comment (AC2)

**Responses to Reviewer #2 of the manuscript "Vertically resolved aerosol variability at the Amazon Tall Tower Observatory under wet season conditions" by Franco & Valiati et al., submitted for publication in Atmospheric Chemistry and Physics**

Dear Editor, we would like to thank reviewer #2 for the valuable comments and useful suggestions to improve our manuscript. Below, you can find answers and actions for each individual comment. In order to make it easier to identify the individual answers and actions, we used the following color code strategy:

● **In black are the reviewer's comments.**
● **In blue are the author's responses.**
● ***In blue and italics are the text modifications we made in the manuscript.***

General comment:

The study discussed vertical variations in size distribution and numerical concentration of submicron aerosols, in addition to intensive and extensive properties of aerosols. These measurements carried out at ATTO are in themselves a great scientific contribution, in addition, the authors discuss these variations satisfactorily with a vast current bibliography, which supports the justification of the study and favors explanations of the observed phenomena.

The authors found significant differences in concentrations and optical properties for different heights and explained these differences using consistent methods, such as analysis of downdraft events and analysis of the aerosol refractive index. Throughout the discussion, several processes of formation, removal, and contributions arising from aerosol transport are highlighted.

The study makes an important scientific contribution that elucidates the processes of emission, formation, and transport of aerosol along the vertical profile. The text is well structured and written, and the figures (main and complementary text) and analysis allow us to reach important conclusions that, in addition to explaining the vertical variations, quantify important parameters such as the absorption of BC, BrC, and optical properties at different heights, which contributes for future model development and tuning in inversion algorithms for satellite products.

I recommend publication after minor revision.

We thank Referee #2 for the very constructive comments and useful suggestions. They helped us to clarify important aspects of our discussions and, thus, to improve the manuscript overall.

Specific comments:

About the Introduction: The text is very well written. The information is clear and objective, and the citations are appropriate and contextualize what has already been done and what is new about the work. In my opinion, the only point that needs to be adjusted is the textual term "vertical distribution of aerosols" which is mentioned in the last paragraph, when highlighting the objective of the work. As this study analyzed measurements (optics, size distribution, and numerical concentration) at two specific heights, the authors should refer to vertical variation and avoid the terms distribution or vertical profile, unless this terminology is introduced in the text for the specific case.

Thanks for pointing this out. The terms "vertical distribution" and "vertical profile" were changed to "vertical variation" in the Introduction.

Section 2.1: Although the authors cite Andreae et al. (2015) for additional information from the study site, I encourage the authors to provide a figure with the location of the site, preferably showing a schematic with the arrangement of instruments for different heights, as well as the average canopy height. The scientific contribution of the study justifies this increase, as the work will serve as a reference for many future studies.

Thanks for the suggestion. The final version included a figure showing the location of the ATTO site in relation to the city of Manaus and the Uatumã River (Figure 1 in the revised manuscript). Regarding the canopy height, according to Lang et al. (2023), the average canopy height at ATTO is around 35 m, which is the number usually mentioned in many previous studies. This information was also added to the text.

Section 2.2: The authors justify not analyzing particles larger than 400 nm due to the limitations of SMPS. However, instrumentation definitions only appear in the next topic. I suggest an inversion of these topics.

Thanks. The "Terminology" section, which includes seasons and modes definitions, was placed after the description of the instruments in the revised manuscript.

Section 2.4: What was the cut section, 2.5? the same question for the aethalometer and MAAP. I believe it is important to add this information to the text. "As both MAAP and aethalometers measure at 637 nm, a comparison between the two was performed to ensure that there were no calibration issues." It would be interesting to add this comparison in the supplementary material.

Thanks for pointing that out. All the optical instruments measured with PM2.5 size cut. We added the following line to the revised manuscript to highlight it:

*All optical instruments were operating with a size cut of 2.5 $\mu m$.*

The supplementary material includes now a figure (Figure S2 of the revised manuscript) comparing both 60m instruments for a week during April 2019, showing a very high correlation ($R^2$ = 0.98, p-value < 0.05) between the MAAP and Aethalometer measurements.

Section 3.2: Wilcoxon rank-sum test, the authors find a significant difference using the Wilcoxon test. Were other tests also evaluated? The authors need to discuss the maximum and minimum variations for these coefficients, as visually there does not appear to be a significant difference for the different heights. I encourage the use of more robust tests using parametric statistics. Although these data do not present a normal distribution, I believe that the sample size justifies the application of parametric statistics.

The Wilcoxon rank-sum test is a commonly used tool to evaluate if two sets of data samples come randomly from the same population. For all properties shown in Section 3.2, the rank-sum test yielded a virtually null p-value, indicating that the distributions are different. Visually, indeed, the difference between the two heights is generally very small, but the notches on the boxplots indicate that the medians are significantly different in all cases. Furthermore, using parametric statistics, this thesis is again supported by comparing the distribution in both heights using the Z-test. For both scattering and absorption coefficients, the Z scores of the distributions at 60 and 325m were well above the significance level of 99% (Z~3), meaning that the distributions are different. The evaluation of these statistical tests was included in the revised manuscript as follows:

*The statistical significance obtained with the Wilcoxon rank-sum test for the absorption coefficient also presented a p-value < 0.01 and a Z-score of the means above the 99\% level threshold, indicating that the differences are statistically significant.*

Section 3.6: Although the study focuses on vertical differences for the real and imaginary components of the aerosol refractive index, and therefore has precise estimates for each height. I ask the authors: is there a possibility of comparing the average value of m with the AERONET inversion estimates? This would be an interesting topic for future work.

Thank you very much. We appreciate this idea for future studies. At the moment, we focused our refractive index analysis on the fine mode component at fixed heights, with calculations made using SMPS size distribution and optical

property measurements. AERONET, on the other hand, uses measurements obtained along the entire vertical profile and considers both fine and coarse aerosol size modes. Therefore, with what was done, there would be no possibility of a direct comparison with AERONET. Still, a future approach that integrates the vertical distribution of aerosols throughout the troposphere and their optical properties could be compared and validated with AERONET.

Technical corrections:

P1, P11, P16, P21, and P22: Avoid the term "slightly".
Example P11: "the absorption coefficients are slightly higher at 325 than at 60 m" replace with the absorption coefficients are greater in magnitude at 325 than at 60 m. Avoid "slightly higher".

Thanks. This was corrected in the text.

References:

Lang, N., Jetz, W., Schindler, K., and Wegner, J. D.: A high-resolution canopy height model of the Earth, Nature Ecology & Evolution, 7, 1778–1789, 2023.